# Deep-Learning-Based COVID-19 Diagnosis and Implementation in Embedded Edge-Computing Device

**DOI:** 10.3390/diagnostics13071329

**Published:** 2023-04-03

**Authors:** Lu Lou, Hong Liang, Zhengxia Wang

**Affiliations:** 1School of Information Science and Engineering, Chongqing Jiaotong University, Chongqing 400074, China; 2School of Computer Science and Technology, Hainan University, Haikou 570100, China

**Keywords:** COVID-19, deep learning, attention mechanism, mixed loss, NVIDIA Jetson devices

## Abstract

The rapid spread of coronavirus disease 2019 (COVID-19) has posed enormous challenges to the global public health system. To deal with the COVID-19 pandemic crisis, the more accurate and convenient diagnosis of patients needs to be developed. This paper proposes a deep-learning-based COVID-19 detection method and evaluates its performance on embedded edge-computing devices. By adding an attention module and mixed loss into the original VGG19 model, the method can effectively reduce the parameters of the model and increase the classification accuracy. The improved model was first trained and tested on the PC X86 GPU platform using a large dataset (COVIDx CT-2A) and a medium dataset (integrated CT scan); the weight parameters of the model were reduced by around six times compared to the original model, but it still approximately achieved 98.80%and 97.84% accuracy, outperforming most existing methods. The trained model was subsequently transferred to embedded NVIDIA Jetson devices (TX2, Nano), where it achieved 97% accuracy at a 0.6−1 FPS inference speed using the NVIDIA TensorRT engine. The experimental results demonstrate that the proposed method is practicable and convenient; it can be used on a low-cost medical edge-computing terminal. The source code is available on GitHub for researchers.

## 1. Introduction

Multiple instances of pneumonia of unknown origin were originally found in Wuhan, Hubei Province, China, in December 2019, and were later proven to be caused by a coronavirus known as Severe Acute Respiratory Syndrome Coronavirus 2 (SARS-CoV-2). The World Health Organization (WHO) then announced the name of the novel lung disease caused by the coronavirus as COVID-19 [1]. Early symptoms of illness caused by the virus include fever, cough, fatigue, and, in some patients, headache, hemoptysis, and diarrhea [2]. Since the transmission route of COVID-19 is via droplets, aerosols, close contact, etc., the virus is highly contagious and can easily cause infection in a wide range of people. As of 7 June 2022, 530 million confirmed cases have been reported to the World Health Organization globally, with 6.3 million deaths [3]. Facing such a severe situation, the early diagnosis and classification of suspected cases are particularly important.

The most commonly used method for detecting SARS-CoV-2 is real-time reverse transcription polymerase chain reaction (RT-PCR). It is regarded as the gold standard for the diagnosis of SARS-CoV-2 due to its high sensitivity and high specificity [4,5], which is achieved by collecting samples from the upper respiratory tract (e.g., nasopharynx or oropharynx). However, this method is time-consuming and requires medical personnel with specialized skills, as well as a specific laboratory [6]. In addition, the RT-PCR method depends on the collected samples, and if the samples are not collected, transported, and stored appropriately, it could lead to false-negative results [7,8]. Another way to detect the disease is by computed tomography (CT) and X-ray, which is achieved by consolidation or ground glass opacity (GGO) in the lungs [9], as indicated by the red rectangular box in Figure 1. While both can be used for the diagnosis of COVID-19, chest CT images can show detailed information about the infected area. Some researchers found that the 98% sensitivity of chest CT is higher than the 71% sensitivity of RT-PCR by comparing CT images and RT-PCR tests at the initial patient visit [10]. This indicates that CT images of the lungs are more useful for the early diagnosis of patients.

In practical applications, deep learning has found wide application across diverse fields, such as autonomous driving, speech recognition, image classification, and fault diagnosis. Deep learning models are capable of adaptively updating themselves based on changes in data without the need for manual intervention. In comparison to traditional machine learning methods, deep learning methods exhibit higher accuracy due to their ability to automatically extract more features. In recent years, deep learning algorithms, and in particular Convolutional Neural Networks (CNN), have demonstrated remarkable capabilities in various medical tasks, such as medical image segmentation, recognition, and classification. Such algorithms have been utilized in computer-aided medical systems to effectively assist doctors in their diagnoses. Notably, several studies have applied these methods to COVID-19 lesion detection, such as transfer-learning-based methods [13,14], improved model-based methods [15,16,17], and integrated model-based methods [18], among others. Additionally, deep learning has been applied in many real-life scenarios, such as autonomous driving and smart homes. Hasani et al. [19] designed software for COVID-19 detection, where users obtain diagnosis results by inputting images. The transplantation of deep learning methods into embedded devices is a novel field, which can also be used for diagnosing COVID-19 disease on portable devices. Some studies have applied deep learning to android apps, where users input an image, and the pre-trained model extracts features, displaying the results on the user interface [20,21]. Furthermore, some studies also have analyzed and tested the performance of CNN models on different types of NVIDIA Jetson modules, and experimental results have demonstrated the potential of executing CNN models on NVIDIA Jetson modules, particularly Jetson TX2, which performs best in image recognition [22,23].

The aforementioned methods demonstrate the potential of deep learning in enhancing the COVID-19 detection efficiency and its practical value for embedded edge-computing devices. However, these research approaches have notable limitations. Researchers have used self-designed simple CNN models instead of established CNN frameworks, resulting in a lack of comparability and persuasive evidence, while neglecting the impact of inadequate training data on detection accuracy. Furthermore, studies on embedded devices have not devised COVID-19-specific networks but instead utilized pre-trained models. To address these issues, this paper proposes an improved VGG19 method. This proposed method effectively distinguishes similarities in lung diseases, improves model accuracy, reduces model parameters, and can be applied to NVIDIA Jetson devices. This study classifies diseases on two different-sized datasets and sorts the images into three categories: normal, pneumonia, and COVID-19. The main contributions of this paper are as follows:We propose an improved VGGNet to increase the detection accuracy by adding a channel attention mechanism;We use a mixed loss function of center loss and cross-entropy loss to train the model to better distinguish different classes of pneumonia;We apply the algorithm to an NVIDIA Jetson (TX2, Nano) to validate the feasibility of the algorithm on the embedded edge-computing platform.

The rest of the paper is described as follows. Section 2 explains some of the relevant work in this area. Section 3 describes in detail the dataset and the methodology. Section 4 analyzes the experimental details as well as the results. Finally, Section 5 concludes the research and lists future works.

## 2. Related Works

Most of the existing deep-learning-based COVID-19 detection methods are based on chest X-ray images or CT images; this paper introduces some state-of-the-art algorithms according to the categories of image datasets.

### 2.1. X-ray Image-Based Methods

Mijwil [24] used machine learning techniques to classify normal, pneumonia, and COVID-19 cases. Machine learning techniques included random forest, logistic regression, naïve Bayes, and support vector machine. The experimental results showed that the support vector machine technique had the best effect and was able to obtain 91.8% accuracy. Ezzoddin et al. [25] utilized the pre-trained DenseNet network in conjunction with the analysis of variance (ANOVA) feature selection method to identify relevant features, which were subsequently input into the light gradient boosting machine (LightGBM) algorithm for COVID-19 case classification. Performance evaluation was conducted on both multi-class and binary classification tasks, resulting in achieved accuracies of 94.22% and 99.20%, respectively. Nasiri et al. [26] leveraged the MobileNet and DenseNet169 network models for COVID-19 diagnosis. The two models were first pre-trained independently and subsequently subjected to the univariate feature selection algorithm to identify optimal features. The light gradient boosting machine (LightGBM) was then deployed for disease classification. Experiments showed that the proposed method achieved accuracies of 98.40% and 91.11% for binary and tri-classification tasks, respectively. Nayak et al. [27] proposed a lightweight CNN model named LW-CORONet to diagnose COVID-19 infection. Compared with existing CNN models, the proposed model requires very few parameters, which is beneficial to reduce computational costs; the model was verified on two datasets, and the multi-classification and binary classification reached 98.67% and 99.00% accuracy on Dataset-1, respectively, and 95.67% and 96.25% accuracy on Dataset-2, respectively. Rahman et al. [28] used several pre-trained Convolutional Neural Networks (CNN) (ResNet18, ResNet50, ResNet101 [29], InceptionV3 [30], DenseNet201 [31], and ChexNet [32]) and UNet segmentation techniques for COVID-19 detection using five different image enhancement techniques, namely histogram equalization (HE), contrast-limited adaptive histogram equalization (CLAHE), image complement, gamma correction, and the balance contrast enhancement technique (BCET), to investigate the effects of image enhancement techniques on COVID-19 detection. The experimental results showed that the accuracy, precision, and recall were 96.29%, 96.28%, and 96.28%, respectively. Khan et al. [17] used two migration-learning-based classification methods for COVID-19 detection and used dropout and regularization techniques to improve the classification performance, and their classification results showed that the proposed method achieved accuracy of 96.13%. Chakraborty et al. [33] used the ResNet18 pre-training model, preprocessing technology, and image segmentation method to predict COVID-19. The accuracy, sensitivity, and specificity obtained on the dataset of X-ray images were 96.43%, 93.68%, and 99%, respectively. Hasani et al. [19] designed software for the automatic detection of COVID-19 using X-ray images, named COV-ADSX, which uses the DenseNet169 network and XGBoost algorithm for detection, and utilizes the Grad-CAM algorithm to show the decision area on a heatmap, which is then presented to the user. The software achieved an accuracy rate of 98.23%, and its portability and visualization aid radiologists in clinical diagnosis. Mahapatra et al. [34] deployed a Deep Convolutional Neural Network in a web application for the detection of chest images. This method tested ResNet50, Inception V3, Xception, and VGG16, and the results showed that Inception V3 was the most accurate in detecting COVID-19, with nearly 99% accuracy in positive cases and 93% accuracy in negative cases.

### 2.2. CT-Image-Based Methods

Chest CT has been proven to be a reliable method of detecting COVID-19. CT images not only have much higher resolution than X-ray images but also can visualize multiple angles of the lesion. Fan et al. [35] proposed a dual-branch COVID-19 detection method based on transformer and CNN. The transformer branch extracts global features and the CNN branch extracts local features, thereby improving the classification accuracy and achieving a classification accuracy score of 96.7%. Ter-Sarkisov [36] designed the COVID-CT-Mask-Net network based on the Mask R-CNN model to detect COVID-19 cases, and verified it on the COVIDx CT dataset, achieving overall accuracy of 95.64%. Yang et al. [37] proposed three integrated deep learning architectures (F-EDNC, FC-EDNC, and O-EDNC), and selected three of the 16 pre-trained models to develop the EDNC architecture. The experimental results show that the F-EDNC method has the best effect and can obtain a 97.55% accuracy rate, while FC-EDNC and O-EDNC achieve 97.14% and 96.32%, respectively. Chetoui et al. [38] fine-tuned and used DenseNet-121 and five EfficientNet models (B0, B2, B3, B4, and B5) for COVID-19 disease detection. Experimental results show that fine-tuning EfficientNet-B0 (CovCTx) achieves the highest accuracy rate of 96.37%. Nair et al. [39] improved ResNet50 and proposed the CORNet model, which was then compared with other network models. This method achieved a recall rate of 90% in COVID-19 detection. Chaudhary et al. [40] proposed a two-stage framework to detect diseases, using a fine-tuned DenseNet framework in the first stage and a fine-tuned EfficientNet architecture in the second stage, and ultimately achieved an accuracy rate of 89.3%. Perumal et al. [41] combined deep learning models (VGG-16, ResNet50, InceptionV3, AlexNet) and machine learning classifiers (SVM, random forest, decision tree, naive Bayes, and K-nearest neighbor) for COVID-19 detection, and they showed that the combination model of AlexNet and SVM obtained the best detection effect, with an accuracy rate of 96.69% and sensitivity of 96%. Garg et al. [42] proposed a three-stage method for diagnosing COVID-19, first using a fine-tuned ResNet50 model for feature extraction, then extracting features for training for different classifications, and finally classifying normal, COVID-19, and CAP, achieving classification accuracy of 95.76%. Maftouni et al. [43] used DenseNet-121 and Residual Attention-92 models to detect CAP and COVID-19 in CT images, pre-trained on the ImageNet and CIFAR-10 datasets, respectively, and finally used a fully connected layer and SVM to output the classification results, achieving an accuracy rate of 95.31%. Verma et al. [20] developed an artificial intelligence (AI)-driven android application for detecting COVID-19 from CT images. Firstly, it segments the lungs in the images and then inputs them into an EfficientNet-B0 model for prediction. If the result is positive, a heat map of the image is outputted. The portability and simple user interface of the android application make it highly practical. Basantwani et al. [21] have also developed an android app that uses a pre-trained InceptionV3 model for transfer learning, enabling users to view COVID-19 diagnoses on the android app.

## 3. Materials and Methods

Figure 2 shows the workflow of the proposed COVID-19 classification method. Firstly, the input images are selected from datasets [11,12,43] and preprocessed using data augmentation and image enhancement. Secondly, the processed images are input into the improved VGG19 to extract image features, and then the classifier is used to classify normal, pneumonia, and COVID-19 cases. Finally, to further illustrate the interpretability of the improved VGG19, the heat map is visualized to highlight the region of interest of the images, which represents the ability of feature extraction for different models.

### 3.1. Dataset Preprocessing

For the data augmentation part, the collected CT images were first rotated randomly and flipped horizontally or vertically, and then were resized by center cropping so that the cropped image resolution was reduced from 512 × 512 to 224 × 224. In the field of image enhancement for CT image processing, the Retinex theory was successfully applied by Zhang et al. [44]. Thus, we used the Multi-Scale Retinex (MSR) enhancement algorithm, which is effective for contrast enhancement and facilitating the observation of details in image lesions [45]. The MSR algorithm performs Gaussian blur on individual scales of the image, subsequently computing the maximum and minimum values through cumulative calculations for each scale, and ultimately linearly quantizing each value. The MSR is calculated as follows: (1)$$R_{MSR_{i}}=\sum_{n=1}^{N}w_{n}R_{n_{i}}=\sum_{n=1}^{N}w_{n}\left[\log I_{i}\left(x,y\right)-\log\left(F_{n}\left(x,y\right)*I_{i}\left(x,y\right)\right)\right]$$
where $R_{MSR_{i}}$ is the *i*th spectral component of the MSR output. *N* is the number of scales, $R_{n_{i}}$ is the *i*th component of the *n*th scale, $w_{n}$ is the weight associated with the *n*th scale, * denotes the convolution operation, $I_{i}\left(x,y\right)$ is the image distribution in the *i*th spectral band, and $F_{n}\left(x,y\right)$ is the surround function.

However, because of the disadvantage that MSR often suffers from “halo” artifacts and low contrast, we first converted the CT images from RGB to grayscale to reduce the amount of computation, and then used the CLAHE [46] method to enhance the contrast of the image, as well as to avoid the loss of image details. Compared to the traditional adaptive histogram equalization (AHE) method, CLAHE is more able to overcome the problems of AHE, such as amplified noise regions in the image, and can better extract the lesion areas. The results of the image preprocessing are shown in Figure 3.

### 3.2. Methodology

For simplicity, we choose the classical Convolutional Neural Network architecture VGG19 as the sample and propose an improved VGG19 network backbone model. The entire network model structure is shown in Figure 4, where the model consists of five stages, and each stage contains 2–4 convolution layers and 1 downsampling layer. In our model, the maximum pooling layer (max-pool) used in VGG19 is turned into a downsampling layer, which consists of convolutional layers to alleviate the information loss problem caused by the max-pool layer, and can better learn the image features [47]. Considering that the traditional classification network that uses the ReLU activation function suffers from the problem of gradient disappearance, our improved VGG19 replaces ReLU with GeLU. In addition, we add the Channel Attention Module (CAM) to replace the last three fully connected layers designed in the original VGG19 network with only one layer, and drastically reduce the weight parameters of the model.

#### 3.2.1. Channel Attention Module

In traditional Convolutional Neural Networks, the extracted feature maps usually contain a lot of redundant information. In order to optimize the learning ability of the network to better extract image features, we make use of CAM [48] to adjust the weight of each channel in the feature map. The CAM structure is shown in Figure 5.

The input feature map is firstly subjected to the average-pooling and max-pooling operations, and two-channel attention vectors are obtained by a multi-layer perceptron (MLP), and then two vectors are combined by element-wise summation. After this, the weights are computed through the sigmoid activation operation and multiplied with the input feature map, and, finally, the channel attention map $M_{c}$ is generated. The channel attention formula is defined as follows: (2)$$M_{c}=\sigma \left(MLP\left(AvgPool\left(F\right)\right)+MLP\left(MaxPool\left(F\right)\right)\right)=\sigma \left(W_{1}\left(W_{0}\left(F_{avg}^{c}\right)\right)+W_{1}\left(W_{0}\left(F_{max}^{c}\right)\right)\right)$$
where *F* represents the input feature map, σ denotes the activation function, $W_{0}$ and $W_{1}$ represent the weight calculation of the MLP on the feature map, $W_{0}\in R^{C/r\times C}$, $W_{1}\in R^{C\times C/r}$, and *r* refers to the reduction rate.

#### 3.2.2. Mixed Loss

The lesion characteristics of COVID-19 and common pneumonia on early CT images of lesions are very similar, which could render the Convolutional Neural Network model unable to classify them. Considering the small distance between categories in the current classification task, the center loss and cross-entropy (CE) loss are used to optimize the model. CE loss is used to calculate the distance between the model predictions and the true labels to guide the model for better classification, but it will not expand the distance between similar classes. Center loss was proposed by Wen et al. [49], and it reduces the intra-class spacing by learning the depth feature centers of each class and optimizing the distance between features and feature centers while increasing the difference between features of different classes. In this paper, multiple loss functions are used to train the model, so that the model can better distinguish between COVID-19 and common pneumonia, thereby improving the screening effect. The formula for each loss function is defined as follows: (3)$$CE\;Loss=-\sum_{i=1}^{n}y_{i}log\left(p_{i}^{c}\right)$$
(4)$$Center\;Loss=\frac{1}{2}\sum_{i=1}^{n}\left|\right|x_{i}-c_{y_{i}}{\left|\right|}_{2}^{2}$$
(5)$$Loss_{all}=CE\;Loss+Center\;Loss$$
where *n* is the number of training samples, $y_{i}$ is the true label of the ith sample, $p_{i}^{c}$ is the predicted probability that sample *i* belongs to category *c*, $x_{i}$ is the feature of the ith sample, and $c_{y}$ denotes the feature center of category *y* in the multi-classification task.

## 4. Experiment and Results

The experiment was conducted in a PC platform where the PC is an Intel(R) Xeon(R) Gold 6330 @ 2 GHz CPU, with 48 GB RAM, and an NVIDIA GTX 3090 with 24 G video memory. In the training process, the Adam optimizer is used to minimize the model loss, and its learning rate is 0.0001, $\beta_{1}$ is 0.9, and $\beta_{2}$ is 0.999. A periodic learning rate decay strategy is also applied, where the decay period is 20 and the decay rate is 0.5. In addition, the batch size is 64 and the epoch is 100.

### 4.1. Dataset

For the training and testing, we used two datasets, COVIDx CT-2 [11,12] and the integrated CT scan dataset [43]. The COVIDx CT-2 dataset has two diverse, large-scale datasets named COVIDx CT-2A and COVIDx CT-2B, and we chose COVIDx CT-2A. The patient cases collected in COVIDx CT-2A come from various organizations around the world, such as the China National Center for Bioinformation (CNCB), the National Institutes of Health Intramural Targeted Anti-COVID-19 (ITAC) Program, the Negin Radiology Medical Center, etc. The COVIDx CT-2A dataset contains 194,922 CT images (60,083 normal CT images, 40,291 images of patients with pneumonia, and 94,548 images of patients with COVID-19) from 3745 patients, with an image size of 512 × 512. In the integrated CT scan dataset, data are obtained from seven public datasets that contain 7593 COVID-19 images from 466 patients, 6893 normal images from 604 patients, and 2618 CAP images from 60 patients. In Figure 6, three categories of CT scan samples are illustrated, namely normal, COVID-19, and pneumonia. Among them, COVID-19 samples show a large number of GGO in the lungs. We divide the dataset into a training set, validation set, and test set in our experiments. For the COVIDx CT-2A dataset, we use 70% of the total samples for training, 10% of the total samples for validation, and 20% of the total samples for testing. For the integrated CT scan dataset, we use 60% of the total samples for training, 20% of the total samples for validation, and 20% of the total samples for testing. The specific distribution of the numbers is shown in Table 1.

### 4.2. Evaluation Metrics

A confusion matrix counts the number of predicted and actual results in the test set and is used to evaluate the performance of the classifier models. In our experiment, the 3 × 3 confusion matrix contains three category classes (0 for normal, 1 for pneumonia, and 2 for COVID-19); when one class (row label) is set to a positive sample, another two classes should be regarded as negative samples. Therefore, we can use binary classification to calculate the evaluation metrics of accuracy, precision, recall, and $F_{1}$-score through the confusion matrices, and the formula of each evaluation metric is defined as follows: (6)$$Accuracy=\frac{TP+TN}{TP+FP+TN+FN}$$
(7)$$Precision=\frac{TP}{TP+FP}$$
(8)$$Recall=\frac{TP}{TP+FN}$$
(9)$$F_{1}-score=2\times \frac{Precision\times Recall}{Precision+Recall}$$
where true positive (TP) indicates the number of positive samples that the model predicted correctly. True negative (TN) indicates the number of negative samples that the model predicted correctly. False positive (FP) indicates the number of samples that the model predicted to be positive but are actually negative, and false negative (FN) indicates the number of samples that the model predicted to be negative but are actually positive.

### 4.3. Results

#### 4.3.1. Preprocessing Comparison

In order to further demonstrate the efficacy of preprocessing, a comparative analysis of preprocessing was conducted on two distinct datasets. Table 2 presents the findings, which indicate that the images processed using the CLAHE and MSR methods exhibit favorable comparisons with the original images. As a result of the variation in the density and configuration of the lung tissue, the brightness and contrast of different regions can differ in the CT images of the lungs. The CT images are usually acquired at a low dose of radiation, which can result in problems with noise and have an effect on the results of the diagnostic procedure. The CLAHE method is capable of enhancing the contrast of lung CT images through adaptive histogram equalization based on local information in different regions. This method effectively mitigates the issue of high contrast by imposing limitations on the range of gray levels for enhanced pixels, while also providing some level of noise suppression. The MSR method exhibits notable flexibility, allowing for the adjustment of parameters to cater to different processing needs. Moreover, it enhances the details present in lung CT images, thereby improving the location of lesion areas within the lung. As shown in Table 2, the utilization of the CLAHE and MSR techniques can enhance the accuracy of disease detection on the integrated CT scan dataset. However, on the COVIDx CT-2A dataset, results indicate that the use of only the CLAHE method is no less effective than using both CLAHE and MSR. We think that when the sample volume of the training dataset is sufficient, the model can learn enough meaningful features without suffering from noise, and therefore the preprocessing methods become less necessary.

#### 4.3.2. Loss Curve

To fully analyze the performance of the model during the training process, we plot the loss function with the horizontal axis representing the number of iterations and the vertical axis representing the training loss. This is shown in Figure 7, where Figure 7a,b, respectively, represent the loss curves of the improved VGG19 on the COVIDx CT-2A dataset and the integrated CT scan dataset, and Figure 7c,d, respectively, represent the loss curves of the original VGG19 on the COVIDx CT-2A dataset and the integrated CT scan dataset.

On the COVIDx CT-2A dataset, the loss value of the original VGG19 on the validation set gradually increases after training more than 40 epochs, meaning that the original VGG19 undergoes overfitting and its generalization ability begins to decrease gradually. In contrast, the improved VGG19 gradually converges after 60 epochs of training and the variance present is smaller than that of the original VGG19, which indicates that the improved VGG19 has better feature extraction and generalization ability than the original VGG19. This demonstrates the effectiveness of introducing the attention mechanism and mixed loss function. During the training process, the total training time of the improved VGG19 is around 14 h, which is longer than the 9 h of the original VGG19 because of its increased computational complexity.

On the integrated CT scan dataset, the loss values of the original VGG19 on the validation set are closer to those on the training set, but the improved VGG19 has lower loss values, and therefore it indicates that the improved VGG19 also has better detection performance on small-scale datasets.

#### 4.3.3. Parameters and Complexity

As shown in Table 3, the weight parameters and operational performance of the improved VGG19 are significant differences compared with the original VGG19. In this paper, since we reduce the three fully connected layers in the original VGG19 to one layer, the number of parameters of the improved VGG19 is reduced by around six times compared with the original model, making the model less prone to overfitting and facilitating the embedded porting of the model. However, the attention mechanism slightly increases the complexity of the model and thus the computational performance becomes slightly higher than in the original VGG19. We test the proposed model on the COVIDx CT-2A and the integrated CT scan dataset with the PC platform, which can achieve 421 fps and 282 fps, respectively. To further evaluate the practicability of the proposed method, we develop the model on an NVIDIA Jetson device, and the model is able to achieve nearly full accuracy of the classification and approximately 0.6−1 FPS of inference speed.

#### 4.3.4. Confusion Matrix and Classification Performance

To comprehensively evaluate the performance of the improved VGG19 and visualize the classification results, we use the confusion matrix to analyze the three-class classification results obtained in the testing phase. In this paper, the improved VGG19 and the original VGG19 model are tested on the COVIDx CT-2A dataset and the integrated CT scan dataset. Figure 8 represents the confusion matrix visualization classification results of the improved VGG19 and the original VGG19 model tested on the COVIDx CT-2A dataset. Figure 9 represents the confusion matrix visualization classification results for the original VGG19 and the improved VGG19 tested on the integrated CT scan dataset.

As shown in Figure 8, the improved VGG19 has 307 misclassified images in the test set of 25,658 images, with an error rate (1-accuracy) of 1.20%, while the original VGG19 has 431 misclassified images, with an error rate of 1.68%. The improved VGG19 can achieve 97.37%, 98.71%, and 99.58% accuracy in the COVID-19, pneumonia, and normal categories, which is around 1% higher than the original VGG19. As shown in Figure 9, the improved VGG19 has 34 misclassified images in the test set of 1572 images, and the error rate is only 2.16%, while the original VGG19 has 109 images misclassified, with an error rate of 6.93%. The improved VGG19 can achieve 96.95%, 99.05%, and 97.50% accuracy in the COVID-19, pneumonia, and normal categories, which is around 1–6% higher than the original VGG19. Overall, the improved VGG19 achieves better accuracy than the original VGG19 in all categories.

#### 4.3.5. Lesion Visualization

To further illustrate the interpretability of the improved VGG19, the Grad-CAM [50] technique is used in this work. First, given an image and a target class as input, the feature map and the predicted values of the class are obtained by the CNN. Next, the predicted values of the target class are back-propagated to obtain the target class gradient information on the feature map, and then the feature map is a weighted summation and the heat map is obtained by ReLu. Finally, the heat map is upsampled to the resolution of the original map by bi-linear interpolation, and then superimposed with the original map to obtain the final visualization result. We randomly selected different categories of CT images (normal, pneumonia, and COVID-19) for validation. As shown in Figure 10, the CT images of COVID-19 and pneumonia have distinctive features, which are significantly different from the normal CT images; it demonstrates the applicability of our method in computer-aided medical diagnosis.

#### 4.3.6. Comparison with State-of-the-Art Methods

To discuss the effectiveness of the proposed method, we performed a contrastive analysis of the two datasets—the COVIDx CT-2A dataset and the medium-sized dataset. First, we tested the COVIDx CT-2A dataset to obtain evaluation metrics and compared it with VGG19 and the state-of-the-art methods proposed by Fan et al. [35], Ter-Sarkisov [36], Yang et al. [37], Hasija et al. [51], and Chetoui et al. [38]. The experimental results depicted in Table 4 demonstrate that the improved VGG19 model outperforms the original VGG19 model in all metrics. The proposed model exhibits higher accuracy in the detection of COVID-19, as evidenced by the confusion matrix presented in Figure 8. The optimization of the improved VGG19 model using the mixed loss function enables the model to better differentiate between COVID-19 and common pneumonia, thereby reducing misclassification in disease detection. It is worth noting that Yang et al. used an integrated model for COVID-19 detection that combines the strengths of multiple neural network models and diagnoses lesion classes in CT images with a recall rate of 100.0%. However, this approach requires a large memory footprint and is not practical for real-world applications. Moreover, the small dataset used in this experiment may lead to overfitting or underfitting. In terms of recall, our proposed model is second only to Yang et al.’s method, indicating that the improved VGG19 model can effectively screen for both common pneumonia and COVID-19 while maintaining a small number of parameters. This makes it more practical, as it can be deployed on embedded devices for COVID-19 detection.

To demonstrate the improved discriminative performance of the VGG19 model, we conducted training and testing on a moderately large dataset and compared it with the original VGG19, Nair et al. [39], Chaudhary et al. [40], Perumal et al. [41], Garg et al. [42], and Maftouni et al. [43], as shown in Table 5. The results presented in Figure 9 indicate that the improved VGG19 outperforms VGG19 in both CAP and COVID-19 categories, achieving a 4.77% higher recall rate. Maftouni et al. proposed a deep ensemble network, which integrates Residual Attention-92 and DenseNet-121 for COVID-19 detection with the FC layer and SVM as the classifier layer. This method exhibited 0.1% higher precision than the modified VGG19, indicating its effectiveness in detecting positive cases. However, the method only performs a dichotomous classification task, which overlooks the need for general pneumonia detection in the diagnosis of lung diseases. As shown in Table 5, our method tests the classification metrics more comprehensively than some state-of-the-art algorithms in terms of performance evaluation, which makes the method more applicable. Overall, the proposed method has significant improvements in all metrics.

#### 4.3.7. Implementation on NVIDIA Jetson Device

In the experiment, we also ported the proposed method to the low-cost NVIDIA Jetson device (TX2, Nano) to evaluate the performance of the model. NVIDIA Jetson devices are embedded development boards developed by NVIDIA. Among them, the Jetson Nano features a 128-core Maxwell GPU, quad-core ARM Cortex-A57 CPU, and 4 GB 64-bit LPDDR4 25.6 GB/s memory; the Jetson TX2 features a 256-core Pascal GPU, quad-core ARM Cortex-A57, and dual-core Denver2 CPU, with 8 GB 128 bit LPDDR4 59.7 GB/s memory. We used Ubuntu 18.04, JetPack 4.6, TensorRT, and a python environment for the Jetson device deployment. NVIDIA has developed a deep learning inference engine, called TensorRT, which can be used to perform inference on models in the NVIDIA Jetson device. TensorRT supports major frameworks such as TensorFlow, PyTorch, Caffe, etc., and can accelerate the inference time for deep learning. This paper uses TensorRT to accelerate inference on the improved VGG19. First, we convert the trained model to onnx format, which can be used for TensorRT inference, and the experimental accuracy is set to fp16. The experimental results show that the inference accuracy of the Jetson device is near to that of the PC platform, approximately 97%, but the inference speed is only 0.6−1 FPS, which is much slower than the PC platform due to the limited hardware resources. However, taking into account the highly cost-effective Jetson devices, they can meet the needs of some practical application scenarios, such as portable edge-computing medical terminals.

## 5. Conclusions

In this paper, a Convolutional Neural Network using VGG19 is proposed as the backbone model for detecting COVID-19 disease in lung CT images. Given the noise and low contrast issues in the original images, we used the CLAHE and MSR methods and conducted experiments that demonstrated the effectiveness of the method in detecting COVID-19 disease. In addition, we used the channel attention module to adjust the channel weights of the feature maps so that the model could focus more on the lesion regions in the CT images to improve model accuracy. The number of model parameters has been lowered by reducing the number of fully connected layers for application to embedded devices. Given the relatively low similarity between disease classes, a mixed loss function was developed to increase the distance between traits in different classes, to minimize the false-positive rate of the model in COVID-19 and pneumonia. Our experiments with both datasets on PC platforms and NVIDIA Jetson devices lead us to conclude that (1) image preprocessing facilitates lung disease classification by the model; (2) increasing the channel attention modulus and mixed loss function can greatly improve the accuracy of the model; (3) model transfer to NVIDIA Jetson devices maintains model accuracy and demonstrates its portability for future development on embedded edge-computing platforms.

However, there are some limitations to this work. In this paper, only three categories of lung CT images were used for classification and detection, and more categories of CT imaging equipment with similar lung diseases should be collected for lung diagnosis in the future. Although the improved VGG19 has fewer parameters than the original VGG19, it is still not lightweight enough to achieve better computing performance. We will consider applying more advance and lightweight deep learning models to accelerate inference on low-cost embedded edge-computing devices.

## Figures and Tables

**Figure 1 diagnostics-13-01329-f001:**
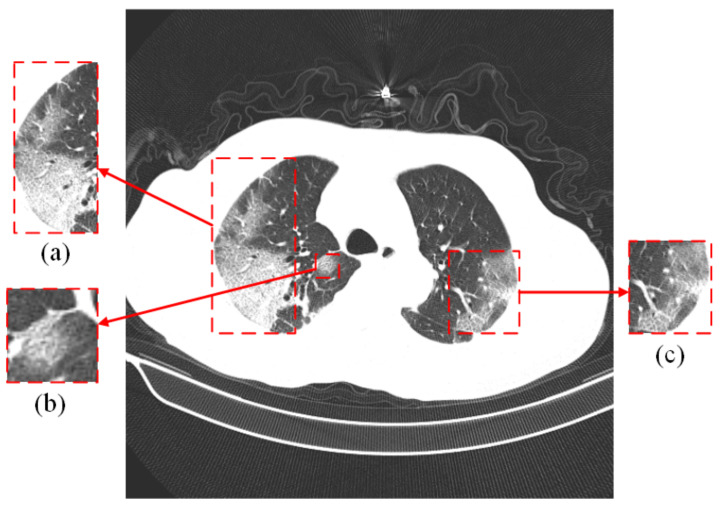
The lung CT image of a COVID-19 patient. (**a**–**c**) are ground glass images. The original image comes from the COVIDx CT-2A dataset [11,12].

**Figure 2 diagnostics-13-01329-f002:**
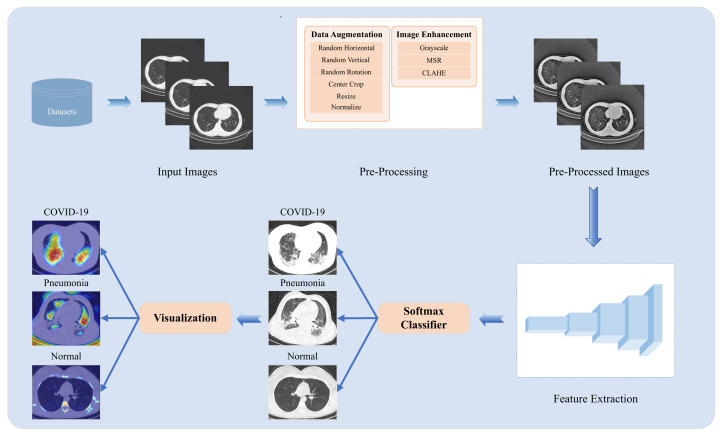
Workflow of the proposed COVID-19 classification method.

**Figure 3 diagnostics-13-01329-f003:**
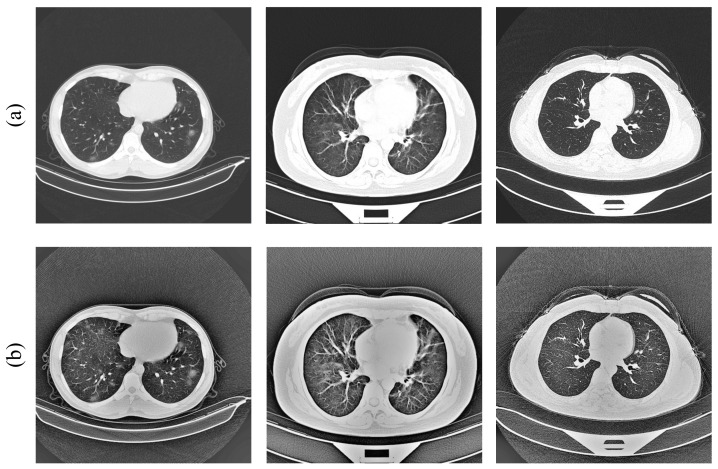
(**a**) Original images; (**b**) processed images [11,12].

**Figure 4 diagnostics-13-01329-f004:**
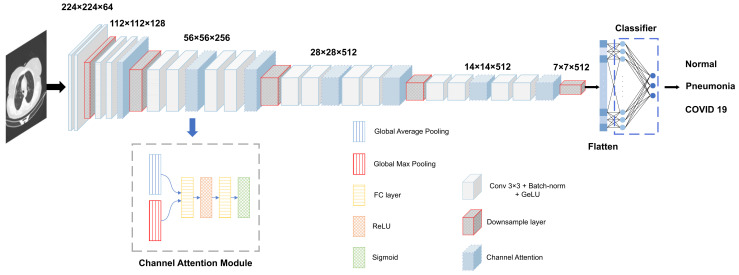
Architecture of the improved VGG19.

**Figure 5 diagnostics-13-01329-f005:**
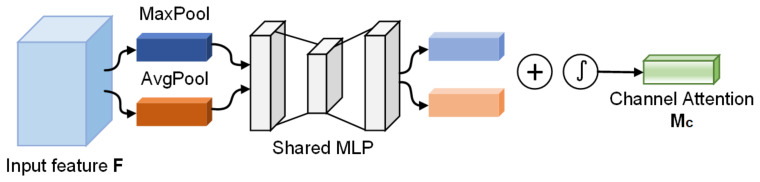
Channel attention module.

**Figure 6 diagnostics-13-01329-f006:**
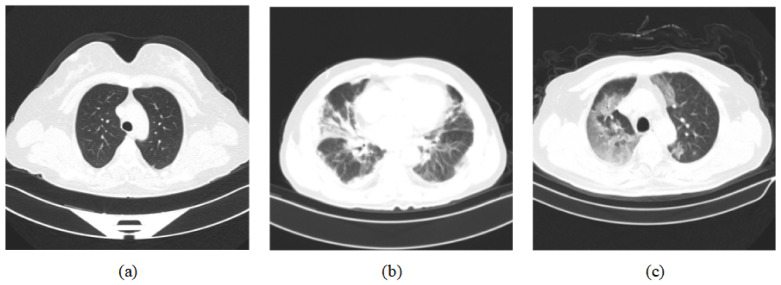
Dataset sample CT images [11,12]. (**a**) represents normal CT scan; (**b**) represents pneumonia CT scan; (**c**) represents COVID-19 CT scan.

**Figure 7 diagnostics-13-01329-f007:**
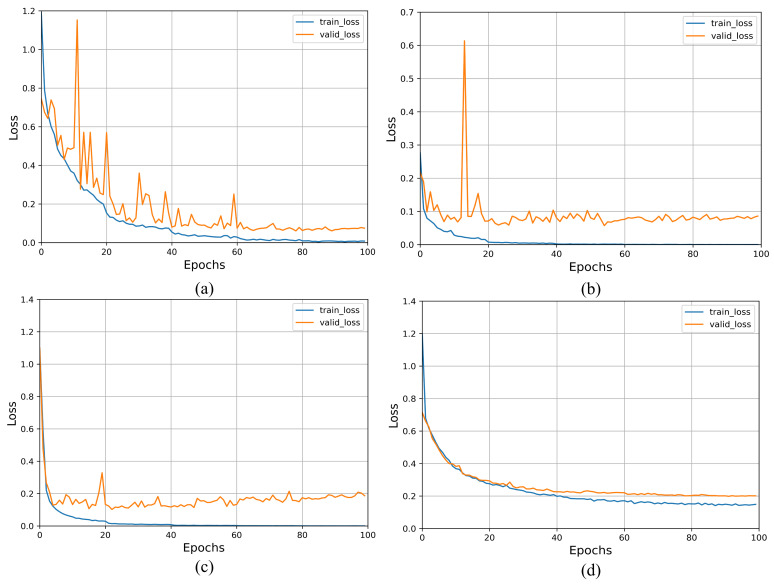
Loss curves of the original VGG19 and the improved VGG19. (**a**,**b**), respectively, represent the loss curves of the improved VGG19 on the COVIDx CT-2A and the integrated CT scan dataset. (**c**,**d**) represent the loss curves of original VGG19 on the COVIDx CT-2A and the integrated CT scan dataset.

**Figure 8 diagnostics-13-01329-f008:**
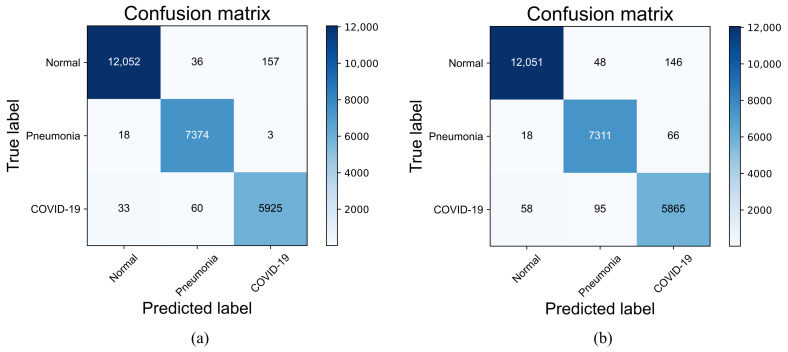
Confusion matrixes of two models tested on the COVIDx CT-2A dataset. (**a**) represents the improved VGG19; (**b**) represents the original VGG19.

**Figure 9 diagnostics-13-01329-f009:**
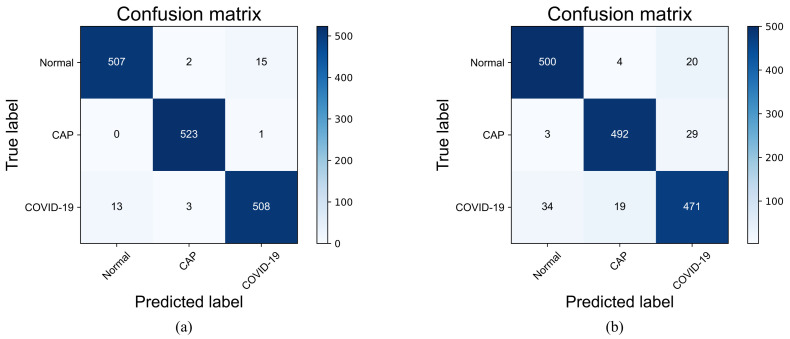
Confusion matrixes of two models tested on the integrated CT scan dataset. (**a**) represents the improved VGG19; (**b**) represents the original VGG19.

**Figure 10 diagnostics-13-01329-f010:**
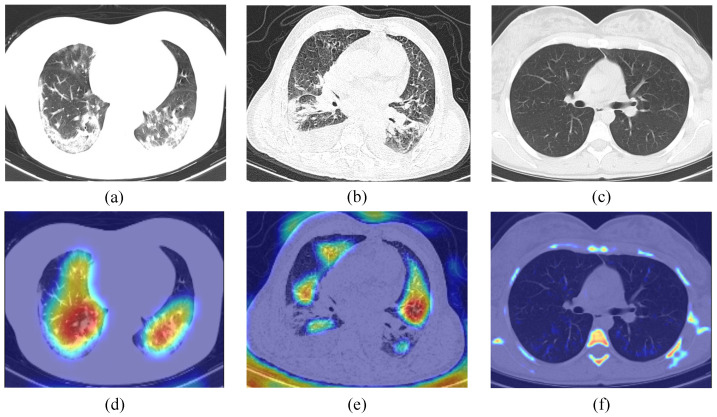
Grad-CAM visualization. (**a**–**c**), respectively, represent COVID-19, pneumonia, and normal CT scan; (**d**–**f**), respectively, represent heatmap of (**a**–**c**).

**Table 1 diagnostics-13-01329-t001:** Data distribution of training set, validation set, and test set.

Dataset	Class	Training	Validation	Test
Integrated CT scan	Normal	1570	524	524
	CAP	1570	524	524
	COVID-19	1570	524	524
	Total	4710	1572	1572
COVIDx CT-2A	Normal	25,496	6244	12,245
	Pneumonia	25,496	6244	7395
	COVID-19	25,496	6244	6018
	Total	76,488	18,732	25,658

**Table 2 diagnostics-13-01329-t002:** Comparison of preprocessing results.

Dataset	CLAHE	MSR	Precision (%)	Recall (%)	F${}_{1}$-Score (%)	Accuracy (%)
Integrated CT scan	✗	✗	97.13	97.14	97.13	97.14
	✓	✗	97.66	97.65	97.65	97.65
	✓	✓	97.83	97.84	97.83	97.84
COVIDx CT-2A	✗	✗	98.39	98.76	98.57	98.66
	✓	✗	98.59	98.83	98.71	98.82
	✓	✓	98.55	98.86	98.70	98.80

**Table 3 diagnostics-13-01329-t003:** Comparison of parameter numbers and complexity of VGG19 and the proposed models.

Model	Params. (M)	Flops (G)
The original VGG19	139.58	19.88
The improved VGG19	21.89	22.30

**Table 4 diagnostics-13-01329-t004:** Comparison of the proposed and state-of-the-art methods on the COVIDx CT-2A dataset.

Methods	Precision (%)	Recall (%)	F${}_{1}$-Score (%)	Accuracy (%)
The original VGG19	97.99	98.25	98.12	98.32
Fan et al. [35]	97.45	97.76	96.36	96.73
Ter-Sarkisov [36]	91.66	-	-	95.64
Yang et al. [37]	95.65	100.0	97.78	97.83
Hasija et al. [51]	98.11	98.06	98.09	98.38
Chetoui et al. [38]	-	97.00	-	96.37
The proposed VGG19	98.55	98.86	98.70	**98.80**

**Table 5 diagnostics-13-01329-t005:** Comparison of the proposed and state-of-the-art methods on the medium-sized dataset.

Methods	Precision (%)	Recall (%)	F${}_{1}$-Score (%)	Accuracy (%)
The original VGG19	93.07	93.07	93.07	93.07
NAIR et al. [39]	-	90.00	-	-
Chaudhary et al. [40]	88.19	87.56	87.87	89.30
Perumal et al. [41]	-	96.00	-	96.69
Garg et al. [42]	-	-	-	95.76
Maftouni et al. [43]	97.93	90.80	94.23	95.31
The proposed VGG19	97.83	97.84	97.83	**97.84**

## Data Availability

The datasets used in this research are publicly available with the name “COVIDx CT-2A | Kaggle” on https://www.kaggle.com/datasets/hgunraj/covidxct, (accessed on 7 June 2022) and the name “Large COVID-19 CT scan slice dataset | Kaggle” on https://www.kaggle.com/datasets/maedemaftouni/large-covid19-ct-slice-dataset, (accessed on 7 June 2022). In addition, the source code is freely available on the GitHub page at https://github.com/One2332x/covid19_clssification, (accessed on 25 October 2022).

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
