# Peer review of "Deep-Learning-Based COVID-19 Diagnosis and Implementation in Embedded Edge-Computing Device"

_diagnostics, 2023, doi:10.3390/diagnostics13071329_

Round 1
Reviewer 1 Report
This manuscript is well and good at innovation and clears the clarity of the reader.
It is well structured and well written. The author does a good job of presenting a highly technical and complicated process in an easy-to-understand manner. Authors need to cross check the reference section by addressing the cited contents in the introduction and related work part. The introduction must be an extended version of the abstract. The authors must elaborate on the points highlighted on the abstract and give supportive ideas and references. The conclusions in this manuscript are primitive. Rewrite your conclusions. References aren’t formatted according to rules.
Additional References:
The following articles could be useful:
Implementation of Machine Learning Techniques for the Classification of Lung X-Ray Images Used to Detect COVID-19 in Humans. https://doi.org/10.24996/ijs.2021.62.6.35
Reviewer 2 Report
The authors present a deep-learning method to detect COVID-19 that has interesting results.
The paper is well-written and organized.
It is important when comparing the proposed VGG19 (Tables 4 and 5), to establish if the classification results are obtained by considering similar classification classes and features.
How does the quality of the original image affect the classification results?
Is there a similar disease, besides those selected in the proposed VGG19, that can affect the classification results?
Why is it important to evaluate the proposed method in embedded platforms?
Reviewer 3 Report
The overall structure of the paper is appropriate, and the manuscript was written quite well. Some comments are provided to improve the quality of this manuscript. In my opinion, this manuscript should be revised before accepting in Diagnostics.
Some comments are provided below:
1) Please consider adding the sources of Fig.1, Fig. 3, and Fig. 4 to their captions to prevent copyright issues.
2) In Fig. 2, in "Feature extraction" part, I think the cubes should be reversed as the extracted features have much lower dimension than the input image.
3) The quality of Fig. 2 and Fig. 5 is very good.
4) Table 1 and Table 2 can be merged into one table. You can add dataset names as a row to the new table.
5) The most recent reference used in "X-ray image-based methods" section is from 2020 and outdated. Please consider adding new references in this section. It is recommended to add the following papers to this section to cover more recent related works:
[1] “Classification of COVID-19 in Chest X-ray Images Using Fusion of Deep Features and LightGBM,” in 2022 IEEE World AI IoT Congress (AIIoT), 2022, pp. 201–206, doi: 0.1109/AIIoT54504.2022.9817375.
[2] “An Efficient Deep Learning Method for Detection of COVID-19 Infection Using Chest X-ray Images,” Diagnostics, vol. 13, no. 1, p. 131, 2023.
[3] “Diagnosis of COVID-19 Cases from Chest X-ray Images Using Deep Neural Network and LightGBM,” in 2022 International Conference on Machine Vision and Image Processing (MVIP), 2022, pp. 1–7, doi: 10.1109/MVIP53647.2022.9738760.
6) Please consider changing "F1-score" to F1-score with one as subscript.
7) Please consider adding a short description of "Multiscale Retinex enhancement algorithm" to the manuscript.
8) The authors compared their proposed method with Refs [39,49,50,51,52,53,54,55,56], but they did not mention these refs in the related works section. I highly recommend to add the description of these works to the related works section.
9) No statistical tests were performed in the paper. So, how could we determine whether the results are statistically significant?
10) The authors should add some real-world applications of AI in Chest X-ray examination software to the related works section of the manuscript as the title of manuscript includes "Implementation in Embedded Edge-computing Device". I highly recommend to check the published papers in Software Impacts (Elsevier) journal and add some published real-world applications of deep learning in detection of COVID-19.
11) Running time of all models should be listed and compared to each other.
12) It is recommended to share a link to the source code to make the project reproducible.
13) Please provide more discussion on the results.
14) I suggest to change the section 5 title to "Conclusion" as it is more common in the research papers.
Round 2
Reviewer 2 Report
The authors have fulfilled the reviewer's comments, and in my opinion, the paper can be published in its present form.
Reviewer 3 Report
In general, the authors have reflected my comments quite well and careful. In my opinion, the idea and experiments are qualified for publishing on Diagnostics.